# Ecohydrological responses to solar radiation changes

**Yiran Wang[1], Naika Meili[2], Simone Fatichi[1]**

[1]Department of Civil and Environmental Engineering, National University of Singapore, Singapore, 117576, Singapore

[2]Singapore-ETH Centre, Future Cities Laboratory Global, Singapore, 138602, Singapore

*Correspondence to*: Yiran Wang(yiranw@u.nus.edu)

**Abstract.** The potential implementation of future geoengineering projects to counteract global warming trends or more generally changes in aerosol loads alter solar radiation reaching the Earth surface. These changes could have effects on ecohydrological systems with impacts which are still poorly quantified. Here, we compute how changes in solar radiation

affect global and local near surface meteorological variables by using CMIP6 model results. Using climate model outputs, we compute climate sensitivities to solar radiation alterations. These sensitivities are then applied to local observations and used to construct two sets of numerical experiments: the first focuses on solar radiation changes only, and the second systematically modifies precipitation, air temperature, specific humidity, and wind speed using the CMIP6 derived sensitivities to radiation changes, i.e., including its land-atmosphere feedback. We use those scenarios as input to a mechanistic ecohydrological model

to quantify the local responses of the energy and water budget as well as vegetation productivity spanning different biomes and climates.

In the absence of land-atmosphere feedback, changes in solar radiation tend to reflect mostly in sensible heat changes, with minor effects on the hydrological cycle and vegetation productivity correlates linearly with changes in solar radiation. When land-atmosphere feedback is included, changes in latent heat and hydrological variables are much more pronounced, mostly

because of the temperature and vapor pressure deficit changes associated with solar radiation changes. Vegetation productivity tends to have an asymmetric response with a considerable decrease in gross primary production to a radiation reduction not accompanied by a similar increase at higher radiation. These results provide important insights on how ecosystems could respond to potential future changes in shortwave radiation including solar geoengineering programs.

## 1 Introduction

Incoming solar (shortwave) radiation is a key variable when studying climate change as it is the main source of continuous energy supply to the Earth (Wild, 2009; Wild et al., 2005). It does not only directly determine the Earth temperature, but also interacts with ecohydrological processes by affecting net radiation and the energy budget at the land surface, as well as the carbon cycle and vegetation dynamics through direct effects on photosynthesis, thus impacting agricultural and natural ecosystems (Comola et al., 2015; Lean & Rind, 1998; Monteith, 1972; Niemeier et al., 2013; Niinemets, 2010; Xia et al., 2014). Over the past 60 years, there have been shifts in solar radiation at the global scale, which have been caused by some minor natural effects of sunspot activity (Lean & Rind, 1998) and mostly by anthropogenic activities (Stanhill & Cohen, 2001; Streets et al., 2006). In North America and Europe from 1950-1980s, a globally decreasing shortwave radiation trend (global dimming) was observed, while shortwave radiation increased back from 1990s onward (global brightening) (Liepert 2002; Wild, 2009). The main reason for the dimming was the surge in aerosol concentrations due to anthropogenic emissions resulting from the rapid industrial development from the middle of the last century to the 1990s (Paasonen et al., 2013; Ruckstuhl et al., 2008), while the brightening since 1990s is due to the anthropogenic control of atmospheric aerosol loads (Wild, 2009; Wild et al., 2005), as well as changes in cloud cover patterns (Pfeifroth et al., 2018; Sanchez-Lorenzo & Wild, 2012). The delayed patterns of dimming and brightening in countries and regions that have experienced a later industrialization and implementations of environmental regulations to limit industrial emissions reinforce these explanations (Manara et al., 2016; Sanchez-Lorenzo & Wild, 2012; K. Wang et al., 2015; Wild et al., 2005). The net effect of these solar radiation changes on ecosystems and ecohydrological variables might be significant, but it has not been quantified, as it is difficult to untangle changes caused by radiation trends alone from the concurrently occurring global warming effects.

In addition to past changes in solar radiation, geoengineering solutions (Caldeira et al., 2013; Irvine et al., 2016) to counteract climate change are often hinged around solar radiation management (SRM). This could be realized by controlling concentrations of aerosols, especially $SO_2$ in the stratosphere (MacMartin et al., 2016), by altering albedo of land and oceans (Irvine et al., 2011) or the cloud cover (Jones et al., 2009), impacting in this way the absorbed energy. For example, albedo can be increased by planting specific plant genotypes with low chlorophyll content (Genesio et al., 2020, 2021) while farming practices, which include the use of no-till management can also increase albedo (Davin et al., 2014). Alternatively, injection of sulphate aerosols into the lower stratosphere can reduce the amount of shortwave radiation reaching the top of the troposphere or placing giant reflectors near the first Lagrange point of the Earth-Sun system can effectively reduce the solar constant (Angel, 2006; Rasch et al., 2008). These solutions are ideated to reduce temperatures and mitigate some of the adverse effects of global warming (Zhang et al., 2015), even though they have been controversial (Barrett et al., 2014; Irvine et al., 2010, 2017) as the consequences of changes in solar radiation on meteorological variables other than temperature and regional climatic patterns could be pronounced. Existing studies suggest that SRM programs are expected to locally stabilize temperatures, but to be unable to revert precipitation changes (Bala et al., 2008; Irvine et al., 2011; K. L. Ricke et al., 2010; Robock et al., 2008; Zhao & Cao, 2022), eventually even exacerbating them (Gertler et al., 2020; K. Ricke et al., 2023). It

emerges that effects of solar radiation management solutions by modifying the amount of shortwave radiation received at the ground they will also alter other meteorological variables as air temperature or precipitation through land-atmospheric feedback in the short run and through climate feedback in the long run. From a process understanding point of view, it is essential to distinguish the impacts of solar radiation itself and that of solar radiation accompanied by land-atmospheric and climate feedback on the local land surface energy budget and ecohydrological response, a topic which has been less studied and frames the scope here. While climate sensitivities to changes in solar radiation are computed using solar-geoengineering simulations – in absence of alternatives – the aim of the study is broader and it is to understand how local hydrological variables and vegetation respond to general alterations in incoming shortwave radiation, which might be caused by specific solar-geoengineering programs or other anthropogenic or natural causes.

To quantify the effects of solar radiation changes from SRM projects on other climate variables, previous studies have examined temperature responses (Bala et al., 2008; Irvine et al., 2011; Kleidon & Renner, 2013), or in a different context, they analyzed the indirect ecohydrological response to solar radiation changes as caused by variability in slope, aspect, and amount of canopy cover (e.g., Zhou et al., 2013; Zou et al., 2007), whereas the direct effect of solar radiation changes on the ecohydrological response have not been analyzed likely due to the complexity of separating the change in solar radiation from changes in temperature and other climatic variables.

Here, we utilize three scenario simulations from the Sixth Coupled Model Intercomparison Project (CMIP6) to isolate as much as possible the effects of a solar radiation changes in absence of temperature change from the overall effect of solar radiation change with its associated land-atmosphere and climate feedback. The first two scenarios correspond to the CMIP6 experiment with abrupt decreased/increased solar radiation (abrupt-solm4p/abrupt-solp4p). The third scenario, the G1 experiment, increased $CO_2$ and reduced solar radiation to maintain a fixed global temperature which helps to isolate the role of solar radiation changes only. These scenarios are used to compute climate sensitivity, i.e., changes in four meteorological variables for a unit of change in solar radiation. Subsequently, we used these sensitivities to construct several numerical experiments aimed at assessing the response of ecohydrological variables to changes in solar radiation with the inclusion (or omission) of land-atmosphere feedback. Specifically, the climate sensitivities derived from the CMIP6 experiments are applied to local meteorological observations and used to run a mechanistic ecohydrological model at the local scale over 115 globally distributed locations corresponding to different biomes and climates. The overall hypothesis is that changes in solar radiation might have considerable implications on the energy and water budgets as well as vegetation productivity, and these effects are amplified when land-atmosphere feedback is included. Furthermore, the numerical experiments provided a mechanistic understanding and interpretation on the spatial heterogeneity of ecohydrological responses to varied solar radiation and its land-atmosphere feedback, which has been difficult to achieve in previous studies.

## 2 Methods and Data

There are at least four ways to study the effects of solar shortwave radiation ($R_{sw}$) changes on the ecohydrology in a given location.

The first is to simply modify incoming shortwave radiation and keep the other meteorological variables unaltered and look at the generated ecohydrological differences. This scenario might be thought to be representative of a very localized geoengineering intervention, but it is unrealistic, as solar radiation changes would induce some changes in other climate variables through local land-atmosphere feedback.

The second option is to include land-atmosphere feedback, in which solar radiation changes lead to a modification of other climate variables, such as locally near surface temperature, precipitation, air humidity, wind speed, but without affecting the overall global climate dynamics, e.g., global temperature is largely unaltered. This intervention might reflect a more regional scale change in aerosol content or a SRM intervention where land-atmosphere feedback takes place or could also be expected as the short-term response to a global scale SRM project.

The third option is to consider all the long-term climate feedback induced by an initial modification of solar radiation. In such a case, global temperature is expected to change in response to a global solar radiation change, with all the associated implications for the climate system. In this third scenario, it is impossible to separate the effects of solar radiation changes from the effects induced by the global temperature change as local land-atmosphere feedback and global climate feedback to changes in solar radiation are overlapped.

Solar geoengineering interventions are aimed at preserving global temperature as $CO_2$ increases. Hence, the fourth scenario is one in which solar radiation effects are isolated from global temperature changes by perturbing two variables (as done in the CMIP6 G1 experiments), usually radiation is reduced, and $CO_2$ is increased to preserve the global scale mean temperature. While $CO_2$ is quite different in this experiment, the changes in $CO_2$ are expected to have a secondary effect on climate, since global temperature, which is the most closely related variable to express overall changes in the climate system (e.g., Seneviratne et al., 2016), remains constant. In this scenario, there is no feedback from a warmer or colder Earth, thus most of the induced changes in climate variables should be directly related to changes in solar radiation and to a minor extent to the different $CO_2$ levels. The experiment (G1) is used in CMIP6 to isolate solar radiation effects, and its results are of high significance to build the no global climate feedback scenario. Such scenario should allow to isolate radiation effects, but still includes some level of land-atmosphere feedback.

In our study, we look in detail at the second and fourth case to understand implications of solar radiation changes on local ecohydrology, but we also report climate sensitivities for the third case. To do so, we first calculated the sensitivity of precipitation, near-surface temperature, specific humidity, and near surface wind speed to changes in surface short-wave downward solar radiation derived from CMIP6 experiments. Second, these calculated climate sensitivities were used to compute local changes in meteorological variables under 10 surface solar radiation perturbation scenarios at 115 globally distributed sites spanning multiple biomes and climate regions (Fig. 1). Specifically, local observed meteorological variables

were perturbed applying the climate sensitivities computed for each location. Third, the Tethys-Chloris (T&C) ecohydrological model was run with the two altered climate forcings (surface solar radiation change only and surface solar radiation change including the land-atmosphere feedback on associated climate variables) to assess the changes in ecohydrological variables to these scenarios (Fig. 1). It has to be noted that the CMIP6 scenarios used here to calculate the climate sensitivities have different

$CO_2$ levels, while we did not perturb $CO_2$ in the T&C experiments as we intended to understand the effects of solar radiation changes, not the overall consequences of a specific geoengineering experiment. The overall workflow of this research is displayed in Fig. 1.

## 2.1 Selection of CMIP6 experiments

We computed solar radiation changes and their associated land-atmosphere and climate feedback for three different scenarios:

1) short-term land-atmosphere feedback ($SR_{sc}$) (second case above, Sect. 2), long-term climate feedback ($SR_{lc}$) (third case above, Sect 2) and no temperature feedback ($SR_{nc}$) (fourth case above, Sect 2). For this purpose, we selected three experiments (G1, abrupt-solm4p, and abrupt-solp4p) from the CMIP6 ensemble (available at https://esgf-node.llnl.gov/search/cmip6/) which perturbated solar radiation, and one control experiment (piControl) as the baseline to assess the response to the solar radiation change. The Cloud Feedback Model Intercomparison Project (CFMIP) provided two of the perturbation experiments

corresponding to an abrupt 4 percent increase (abrupt-solp4p) or decrease (abrupt-solm4p) of the solar constant. The Geoengineering Model Intercomparison Project (GeoMIP) provided one additional experiment where global scale temperature is preserved (the G1 experiment). The G1 experiment includes an abrupt quadrupling of $CO_2$ plus a reduction in total solar constant to maintain a global temperature aligned with the baseline experiment. This scenario without trends in global temperature represents a climate in equilibrium, and while the different $CO_2$ concentration in comparison to the present climate

has some effect on the changes of climatic variables, most of the induced changes should be directly associated to radiation changes in this experiment. We screened all the General Circulation Models (GCMs) and found that only six models can be used for climate sensitivity calculations as they provide the experimental results listed in Table 1. The detailed information of the six GCMs is provided in Table 1 and Table S1. The common period across all the models and experiments spans the hundred years from Jan.1850 to Dec.1949.

**Table 1.** List of models and experiments and associated spatial resolution selected for the climatic sensitivity calculations. NA denotes the model has no available output for the specific experiment.

| Model | piControl | abrupt-solm4p | abrupt-solp4p | G1 |
| --- | --- | --- | --- | --- |
| IPSL-CM6A-LR | 250 km | 250 km | 250 km | 250 km |
| CESM2-WACCM | 100 km | NA | NA | 100 km |
| CNRM-ESM2-1 | 250 km | NA | NA | 250 km |
| MIROC-ES2H | 250 km | NA | NA | 250 km |
| MRI-ESM2-0 | 100 km | 100 km | 100 km | NA |

| CESM2 | 100 km | 100 km | 100 km | NA |

## 2.2 Climate sensitivity calculations based on CMIP6 experiments

We calculated the differences in annual mean values of four climate variables – precipitation, near-surface temperature, specific humidity, and near surface wind speed – plus surface solar radiation for the three experiments (abrupt-solp4p, abrupt-solm4p, G1) and the control conditions (piControl) using six GCMs. We used four models (IPSL-CM6A-LR, CESM2-WACCM, CNRM-ESM2-1 and MIROC-ES2H) to compute the sensitivities for the $SR_{nc}$ scenario with G1 experiments results and three models (IPSL-CM6A-LR, MRI-ESM2-0 and CESM2) to compute the sensitivities for the $SR_{sc}$ scenarios with abrupt-solm4p/abrupt-solp4p experiments results. IPSL-CM6A-LR happened to have both the experiments for $SR_{nc}$ (G1) and $SR_{sc}$ (abrupt-solm4p/abrupt-solp4p) scenarios. The slopes of the linear regressions between annual mean changes in meteorological variables and surface solar radiation were defined as the climatic sensitivity to surface solar radiation changes. Variability in climate sensitivity among the different CMIP6 is accounted for by calculating the sensitivity based on the slope of the linear regression between outputs of the different models, so that uncertainty originated by specific climate model is smoothed. We also use single values of climate sensitivities at the annual scale, rather than monthly or seasonal variable sensitivities to include more data in the computation. However, annual sensitivities tend to show the best correlation with summer sensitivities and the lowest with winter sensitivities. Summer sensitivities are the most relevant to understand ecohydrological changes during the growing season (Fig. S1-S2).

The short-term land-atmosphere and long-term climate sensitivities were calculated using the abrupt-solp4p and abrupt-solm4p scenarios as they integrate the bidirectional changes in solar radiation. Specifically, short-term sensitivities $SR_{sc}$ were computed over the first decade (Jan.1850-Dec.1859), where the global temperature had not yet changed significantly, ten years are selected as a compromise as we needed enough years to average internal climate variability (and remove the uncertainty associated with the selection of one specific year) but not to many to have significant global temperature changes. The long-term sensitivities $SR_{lc}$ were computed using the last 50 years (Jan.1900-Dec.1949) and are thus representative of a different global temperature which impacts the overall Earth climate. The length of 50 years was also chosen to minimize the uncertainty associated with internal climate variability. Notably, because the solar radiation changes in G1 were unidirectional and there were only four models available, we set the intercept as zero (no change expected for no radiation change) to obtain the linear regression slope. In this case we computed the climate sensitivity $SR_{nc}$ using the whole reference period (Jan.1850-Dec.1949) as this simulation is representative of a stationary climate with a constant global temperature.

## 2.3 T&C model

We used the Tethys-Chloris (T&C) model to gain a deeper understanding of the ecosystem response to solar radiation changes with and without the associated land-atmosphere feedback as CMIP6 models do not resolve ecohydrological processes in detail and coarsely parametrize vegetation properties. The mechanistic ecohydrological T&C model is designed for hourly simulations of energy, water, and vegetation dynamics across diverse environments and climates. The model incorporates all

key components of the hydrological cycle and accounts for soil and vegetation heterogeneity. Shortwave and longwave incoming radiation fluxes are explicitly transferred through the vegetation canopy (Ivanov et al., 2008; Wang, 2003). The energy, water and carbon exchanges between the surface (soil and vegetation) and the planetary boundary layer are computed with a resistance analogy scheme (Sellers et al., 1997) accounting for aerodynamic, under canopy and leaf boundary layer resistances, as well as for stomatal, soil-to-root and soil-to-air resistances. The T&C model accounts for vertical soil water content dynamics using the Richards equation. It includes snowpack dynamics and runoff generation mechanisms. Photosynthesis is simulated using the Farquhar biochemical model (Bonan et al., 2011; Farquhar et al., 1980), with a "two big leaves" scheme for net assimilation and stomatal resistance which is simulated using a modified Leuning model (Wang & Leuning, 1998). The model dynamically simulates seven carbon pools, accounting for tissue growth, maintenance respiration, and tissue turnover influenced by environmental stresses. Carbon allocation considers resource availability and allometric constraints, with the ability to translocate reserves for leaf expansion or recovery after disturbances. Phenology is simulated with four growth states, transitioning between states is based on root zone soil temperature, soil moisture, and photoperiod length. The T&C model has been shown to reproduce well energy, water and carbon fluxes at annual, hourly, and seasonal scales as observed from flux towers, as well as it can reproduce other ecohydrological variables such as soil moisture and streamflow. Validation has taken place in all of the 115 globally distributed locations used here, and for most sites is reported in previous studies (Botter et al., 2021; Fatichi et al., 2012a, 2012b, 2014, 2016; Fatichi & Ivanov, 2014; Manoli et al., 2018; Marchionni et al., 2020; Mastrotheodoros et al., 2017, 2020; Meili et al., 2024; Paschalis et al., 2020). For further information on the model's process description and parameterizations, we refer the reader to previous publications (e.g., Fatichi et al., 2012a, 2012b; Fatichi & Pappas, 2017; ;; Paschalis et al., 2022, 2024; Wang et al., 2023; Luo et al., 2024).

## 2.4 Ecohydrological responses to solar radiation changes with T&C modelling

We applied the same linear regression method (used in Section 2.2) to each grid cell in the global CMIP6 simulations and calculated the slope of linear regression as the climate sensitivity of each grid cell under $SR_{sc}$ (short-term land-atmosphere feedback) and $SR_{nc}$ (no temperature feedback) scenarios. Then we selected the closest pixels to the locations of 115 globally distributed sites characterized by different biomes where the T&C model has been tested and used in earlier studies (Section 2.3). These climate sensitivities are then used to perturb the local meteorological observations, which are covering a period between 2 and 39 years, depending on the location. A detailed list of the sites and simulation length is available in Table S2. Specifically, we used 10 levels of solar radiation perturbation plus a control scenario without any solar radiation change for the T&C simulations. The 10 scenarios perturb solar radiation by $\pm 1$ W m$^{-2}$, $\pm 3$ W m$^{-2}$, $\pm 5$ W m$^{-2}$, $\pm 10$ W m$^{-2}$, $\pm 15$ W m$^{-2}$ at the 115 sites respectively and use the derived local climate sensitivities to also modify precipitation, near-surface temperature, specific humidity, and near surface wind speed. These magnitudes of $R_{sw}$ change correspond to reference $R_{sw}$ variations as obtained in global geoengineering studies (see section 3.1). The perturbed meteorological variables are used as T&C model forcing to simulate the associated ecohydrological response at the land surface. The length of the simulation period remains

the same for the control and the perturbed scenarios and it is a function of local data availability (Table S2). All the other

forcing variables (including $CO_2$) and boundary conditions are also unchanged.

The ecohydrological response to $R_{sw}$ changes was assessed by analyzing changes in the land surface energy and hydrological fluxes looking at the different terms of the energy (Eq. 1) and water balance (Eq. 2).

$$R_n = H + \lambda E + G \qquad\qquad (1)$$

$$PR = ET + LK + R \qquad\qquad (2)$$

where $R_n$ represents net radiation, H is the sensible heat flux, $\lambda E$ is the latent heat flux, G is the ground heat flux, PR is the precipitation, ET is the evapotranspiration, LK is the leakage at the bottom of the soil column, and R is surface runoff. We also computed the Bowen ratio ($B_R$) to analyze how changes in $R_n$ are partitioned into changes in H and $\lambda E$. We further analyze the variations in gross primary production (GPP) and leaf area index (LAI) as exemplary variables for vegetation response.

Since the 115 sites exhibit large heterogeneity in climate and biomes, we categorized the 115 sites based on two classification criteria. The first categorization is based on the biome itself which classified the 115 sites into 10 categories (i.e., C3 Grassland, Mixed C3 / C4 Grassland, Evergreen Forest, Tropical Forest, Deciduous Forest, C3 Grassland / Shrubs, C4 Grassland, Savanna, Mixed Forest and Shrubs). The second categorization is based on the wetness index (e.g., Paschalis et al., 2021), i.e., the ratio between precipitation and potential evapotranspiration, computed as PR/PET, sometimes also called aridity index (Arora,

2002). We categorized the 115 sites into three wetness index WI categories: dry (WI $\leq$ 0.5), intermediate (0.5 < WI $\leq$ 1) and wet (WI > 1). The detailed information about these classifications is reported in Table S3 and Fig. S3-S4.

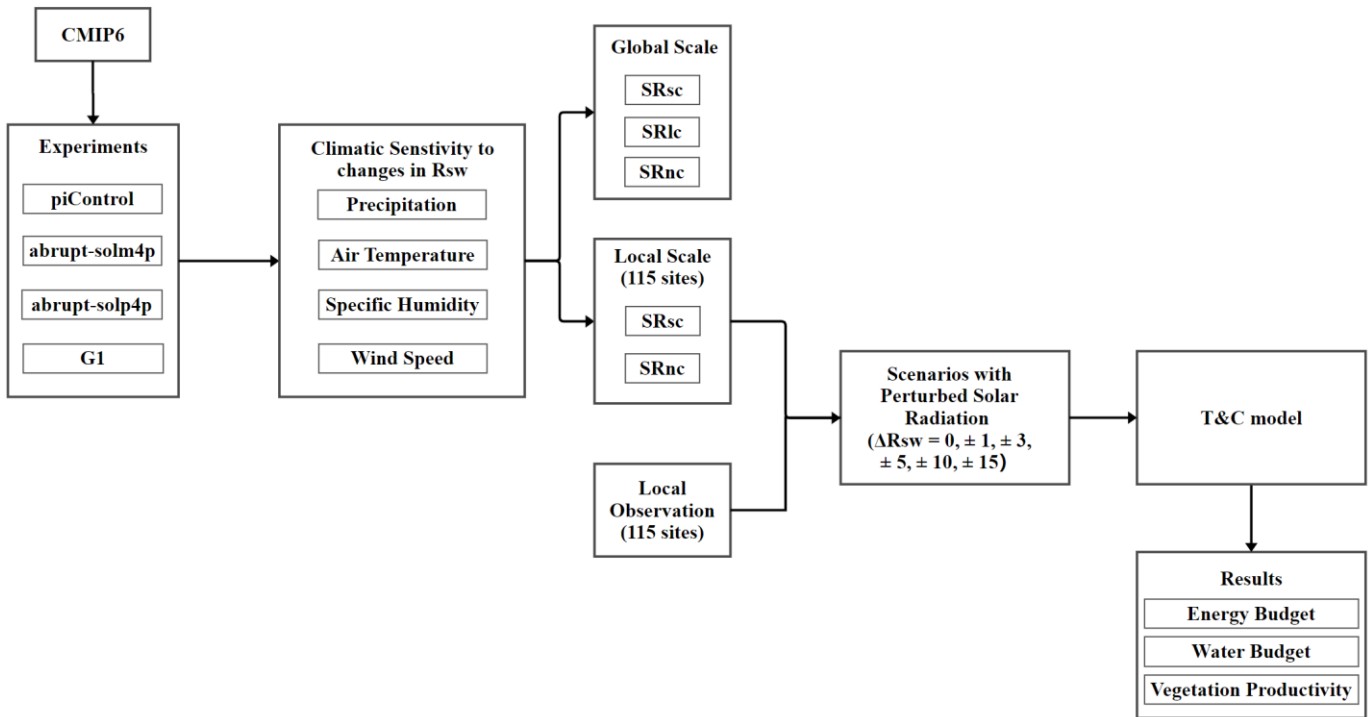

**Figure 1.** Research workflow to illustrate the used data and methodology and performed simulations.

## 3 Results

### 3.1 Climatic sensitivity to solar radiation changes

In agreement with previous studies (Laakso et al., 2020; Russak, 2009; Stanhill, 2011), changes in the analyzed meteorological variables exhibit a positive correlation with changes in surface solar radiation in scenarios involving the land-atmosphere and climate feedback ($SR_{sc}$, $SR_{lc}$ in Fig. 2). In most parts of the world, as expected, a global scale increase in surface radiation leads to an increase in the amount of energy absorbed by the Earth surface, resulting in an increase in surface and air temperature, which in turn increases the specific humidity of the air, as a corollary of the Clausius-Clapeyron relation, and leads to enhanced precipitation (Schneider et al., 2010; Stephens & Ellis, 2008). Changes in wind speeds are relatively small and likely related to enhanced turbulent exchanges or shifts in circulation patterns (Stephens & Ellis, 2008). For most locations on Earth, the changes in surface solar radiation are of the same sign as the changes in the solar perturbation at the top of the atmosphere. However, there are a few regions where the trend of surface radiation changes is opposite to that of the top of atmosphere, which may be due to the complex climate patterns impacting cloud distribution resulting in non-uniform changes in surface solar radiation (Fig. S5).

As expected, sensitivities of precipitation, near surface temperature and specific humidity to changes in $R_{sw}$ are more pronounced when the long-term climate feedback is accounted for than when only the short-term land-atmosphere feedback is considered. In the long-term, the sensitivity of temperature and specific humidity to $R_{sw}$ is even larger than twice the short-term sensitivity (Fig. 2b, 2c). Changes in wind speed are not substantially affected by the climate feedback, with a sensitivity of 0.006 m s-1 per W m$^{-2}$ in both the long and short-term scenarios (Fig. 2d).

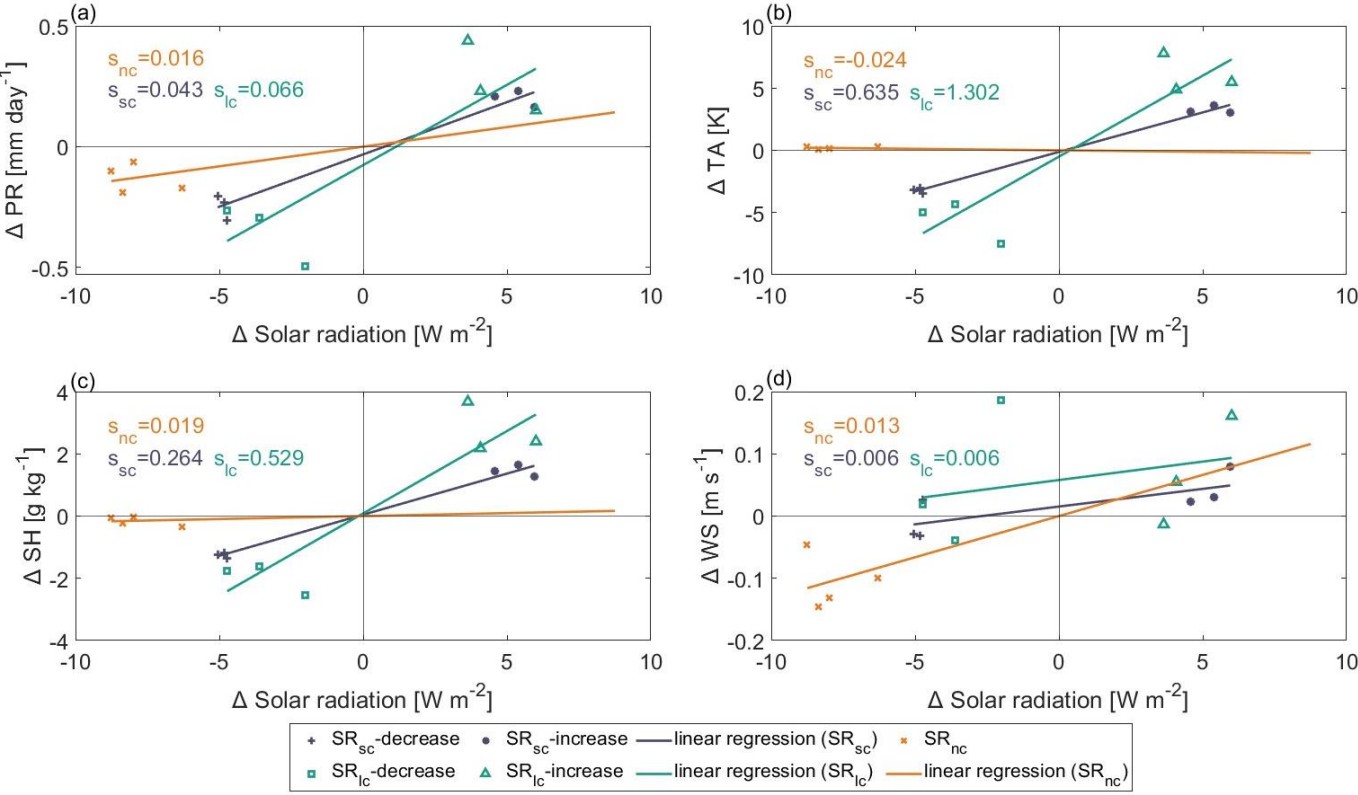

**Figure 2.** Global-scale (over land areas) sensitivity to changes in surface solar radiation of four climatic variables (a) precipitation, (b) near surface temperature, (c) near surface specific humidity, and (d) near surface wind speed. The scattered points indicate global annual means of changes over land between the control scenario (piControl) and three CMIP6 scenarios where solar radiation has been perturbed: G1 (crosses), abrupt-solm4p (plus signs for short-term and squares for long-term changes) and abrupt-solp4p (dots for short-term and triangles for long-term changes) calculated with the models listed in Table 1. The term s denotes the sensitivity of a given climatic variable, which is calculated as the slope of the fitted linear regression. The subscripts denote the sensitivities under the three different conditions: $SR_{nc}$ in orange (no temperature feedback), $SR_{sc}$ in blue (short-term land-atmosphere feedback) and $SR_{lc}$ in green (long-term climate feedback), respectively.

As changes in temperature lead to various dynamics that exacerbate changes in atmospheric patterns, the sensitivity of meteorological variables to surface solar radiation change is greatly reduced when climatic feedback induced by a change in global temperature are excluded, especially the sensitivity of temperature was reduced by an order of magnitude from $SR_{sc}$ to $SR_{nc}$ (from 0.635 K m$^2$ W$^{-1}$ to -0.024 K m$^2$ W$^{-1}$, Table 2). The only exception is wind speed, for which sensitivities remain very small (1% of the mean). This also suggests that global temperature may not be the main driver of wind speed variations when compared to solar radiation. At global scale precipitation and wind speed remain positively correlated with changes in solar radiation under the $SR_{nc}$ scenario, while temperature and specific humidity remain largely unchanged with the slopes of the linear regressions close to zero as derived from the G1 CMIP6 scenario. The spatial distribution of climate sensitivity to solar radiation changes shows remarkable spatial heterogeneity in the short-term ($SR_{sc}$) (Fig. S6a-d) and long-term ($SR_{lc}$) (Fig.

S6e-h) when climate feedbacks are included, while the $SR_{nc}$ scenarios (Fig. S6i-l) show more pronounced latitudinal zonation than the other scenarios.

**Table 2.** Climatic sensitivities to solar radiation changes over global land for the three different conditions: short-term land-atmosphere feedback ($SR_{sc}$), long-term climate feedback ($SR_{lc}$), and no global temperature feedback ($SR_{nc}$).

| Variables [Units] | $SR_{sc}$ | $SR_{lc}$ | $SR_{nc}$ |
|---|---|---|---|
| Precipitation [mm day$^{-1}$ m$^2$ W$^{-1}$] | 0.043 | 0.066 | 0.016 |
| Temperature [K m$^2$ W$^{-1}$] | 0.635 | 1.302 | -0.024 |
| Specific Humidity [g kg$^{-1}$ m$^2$ W$^{-1}$] | 0.264 | 0.529 | 0.019 |
| Wind speed [m s$^{-1}$ m$^2$ W$^{-1}$] | 0.006 | 0.006 | 0.013 |

To better illustrate the global representativeness of the 115 sites selected for the ecohydrological simulations, we compared the distribution of climatic sensitivities computed for these 115 sites with the global distribution of climate sensitivities over land areas from CMIP6 (Fig. 3). Overall, the distribution of the climatic sensitivities for the analyzed sites are in the range of the CMIP6 global distribution of sensitivities, even though the median of the precipitation sensitivities for the 115 sites was slightly lower than the global land median under the $SR_{sc}$ scenario. This is likely due to the fact that the selected locations for which we had model set-ups were mostly positioned in the northern mid-high latitudes (Fig. S3-S4), such as Europe and USA. These regions exhibit lower precipitation sensitivities under the $SR_{sc}$ scenario (Fig. S6a). Nevertheless, the median of climatic sensitivities for the selected 115 sites is still close to the global terrestrial median, and their variance is larger than the global distribution (Fig. 3). Therefore, we conclude that they are fairly representative of the global picture. The variance of the sensitivity distribution increases in the scenarios with long-term climate feedback ($SR_{lc}$) (Fig. 3), which is expected because atmospheric dynamic changes associated with global mean temperature change compound the changes induced by solar radiation.

To select a reasonable magnitude of $R_{sw}$ perturbations for the simulations with the T&C model, we also compare the distribution of solar radiation changes in the 115 sites with the global land distribution obtained from CMIP6 (Fig. S5). The distribution of $R_{sw}$ changes for the selected locations and CMIP6 global land are similar under the $SR_{nc}$ and $SR_{sc}$ scenarios. The range of solar radiation change was around -16 W m$^{-2}$ to 5 W m$^{-2}$ under the $SR_{nc}$ scenario and had a wider range from -18 W m$^{-2}$ to 21 W m$^{-2}$ in the $SR_{sc}$ scenario for the selected locations. Hence, we chose to perturb solar radiation in the range of -15 W m$^{-2}$ to 15 W m$^{-2}$ in the ecohydrological simulations which represents a realistic $R_{sw}$ range consistent with expected local changes in solar radiation from the CMIP6 geoengineering experiments.

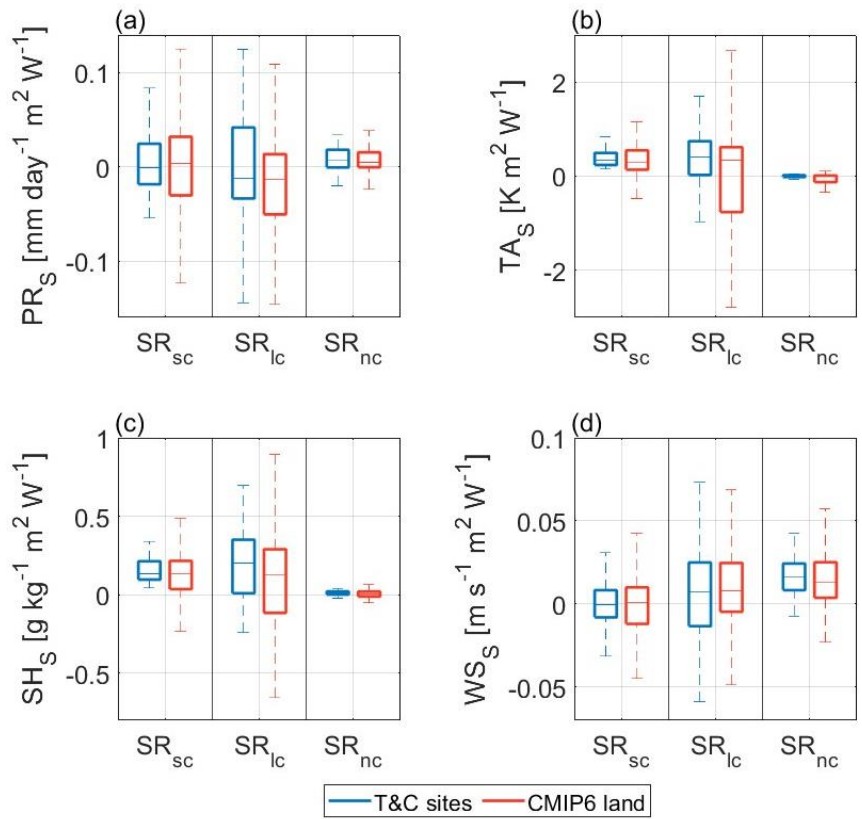

**Figure 3.** Distribution of the climatic sensitivity to a unit change in surface solar radiation for the 115 sites used for the T&C simulations (blue boxes) and global CMIP6 simulations over land (red boxes), for (a) precipitation, (b) near surface temperature, (c) near-surface specific humidity, and (d) near-surface wind speed, under the three different cases: short-term land-atmosphere feedback ($SR_{sc}$), long-term climate feedback ($SR_{lc}$), and no global temperature feedback ($SR_{nc}$).

### 3.2 Solar radiation changes – effects on the energy budget

As expected, an increase/decrease in $R_{sw}$ has a direct impact on the net radiation. For the case without land-atmosphere feedback, the change in $R_{sw}$ translates almost perfectly into a change in $R_n$, (Fig. 4g) with a linear pattern where the change in $R_n$ is about 75% of the change in $R_{sw}$ in all biomes, which roughly corresponds to absorbed $R_{sw}$ (e.g., $R_{sw} \cdot (1\text{-albedo})$). Changes in $R_n$ in the presence of short-term land-atmosphere feedback are slightly more complex and tend to be non-linear for changes in $R_{sw}$ larger than 10 W m$^{-2}$ (Fig. 4a). Consistent with energy conservation, the $R_n$ increase contributes simultaneously to an increase in H and λE even though with different magnitudes. Due to the simulations spanning multiple years, changes in G are relatively modest even in the most extreme $R_{sw}$ perturbations (Fig. S7) and generally much less than 1 W m$^{-2}$, which is an order of magnitude less than changes of H and λE. The extent to which the additional energy in $R_n$ is allocated to H or λE differs considerably between the case with and without land-atmosphere feedback. In the $SR_{sc}$ scenario, there was a greater transfer of heat into λE than into H. The mean change in λE and H is 72% and 28% of the change in $R_{sw}$, respectively, in the simulation

with +5 W m$^{-2}$, and the overall difference is quite pronounced with a mean change in $\lambda E$ of 3.6 W m$^{-2}$ and in H of 1.4 W m$^{-2}$ in the simulation with +5 W m$^{-2}$ $R_{sw}$. As $R_{sw}$ increases, the change in mean $B_R$ across sites is always positive but first decreases and then increases at very high radiation loads ($R_{sw}$ increases larger than 5 W m$^{-2}$), indicating that the energy is firstly allocated proportionally more to $\lambda E$ and then to H, which also suggests that water limitations might start to play a role at very high radiation loads.

In the absence of pronounced land-atmosphere feedback ($SR_{nc}$ scenario), the additional $R_n$ is transferred much more into H. The mean change in $\lambda E$ and H is 17% and 56%, respectively, of the change in $R_{sw}$ in the simulation with +5 W m$^{-2}$ . The extent of the mean change (all subsequent results are computed over the same range from -15 W m$^{-2}$ to 15 W m$^{-2}$ if not specified differently) in H (from -8.2 W m$^{-2}$ to 8.6 W m$^{-2}$, Fig. 4h) was more than double than the mean change in $\lambda E$ (from -3.0 W m$^{-2}$ to 2.4 W m$^{-2}$, Fig. 4i). The variance of changes in the energy budget variables is greater under the $SR_{sc}$ than $SR_{nc}$ scenarios, again showing how land-atmospheric feedback can modify the energy budget at the land surface, beyond the direct effect of a change in solar radiation.

Although $R_n$, H and $\lambda E$ of the different biomes are all positively correlated with changes in solar radiation, sensitivities (computed as a linear change in a given variable as $R_{sw}$ changes from -5 W m$^{-2}$ to 5 W m$^{-2}$, Fig. S8 and Table S4) still varied among biomes. Evergreen forests had the highest $R_n$ and H sensitivities while deciduous and mixed forests had high $R_n$ and $\lambda E$ sensitivities under the $SR_{sc}$ scenario. Tropical forests had the highest $R_n$ and $\lambda E$ sensitivities under the $SR_{nc}$ scenario while the other biomes showed comparable $R_n$ sensitivities which predominantly allocated energy into H in the same scenario. C3/C4 grassland and mixed savanna had the lowest $R_n$ sensitivities under $SR_{nc}$ and $SR_{sc}$ scenarios, respectively. In the $SR_{nc}$ scenario, C3 grassland/shrubs had the highest H sensitivity and the lowest $\lambda E$ sensitivity. In the $SR_{sc}$ scenario, C3 grassland, deciduous forest, C3 grassland/shrubs, C4 grassland, mixed savanna, mixed forest, and shrubs show a decreasing $B_R$ with increasing $R_{sw}$ (negative sensitivity), whereas evergreen forest, C3 grassland/shrubs and C3/C4 grassland have positive $B_R$ sensitivity, which points to some potential water limitation effects. The sensitivity to changes in $R_{sw}$ grouped by wetness index categories differed minimally except for the patterns in $\lambda E$ and $B_R$ in the $SR_{sc}$ scenario, which showed a decreasing $B_R$ and proportionally more $\lambda E$ in the wet locations, while $B_R$ sensitivities are negative in intermediate and dry sites, pointing to water limitations likely induced by changes in precipitation patterns and temperature rather than changes in solar radiation alone as the $SR_{nc}$ scenario does not show any difference across wetness conditions (Fig. S9).

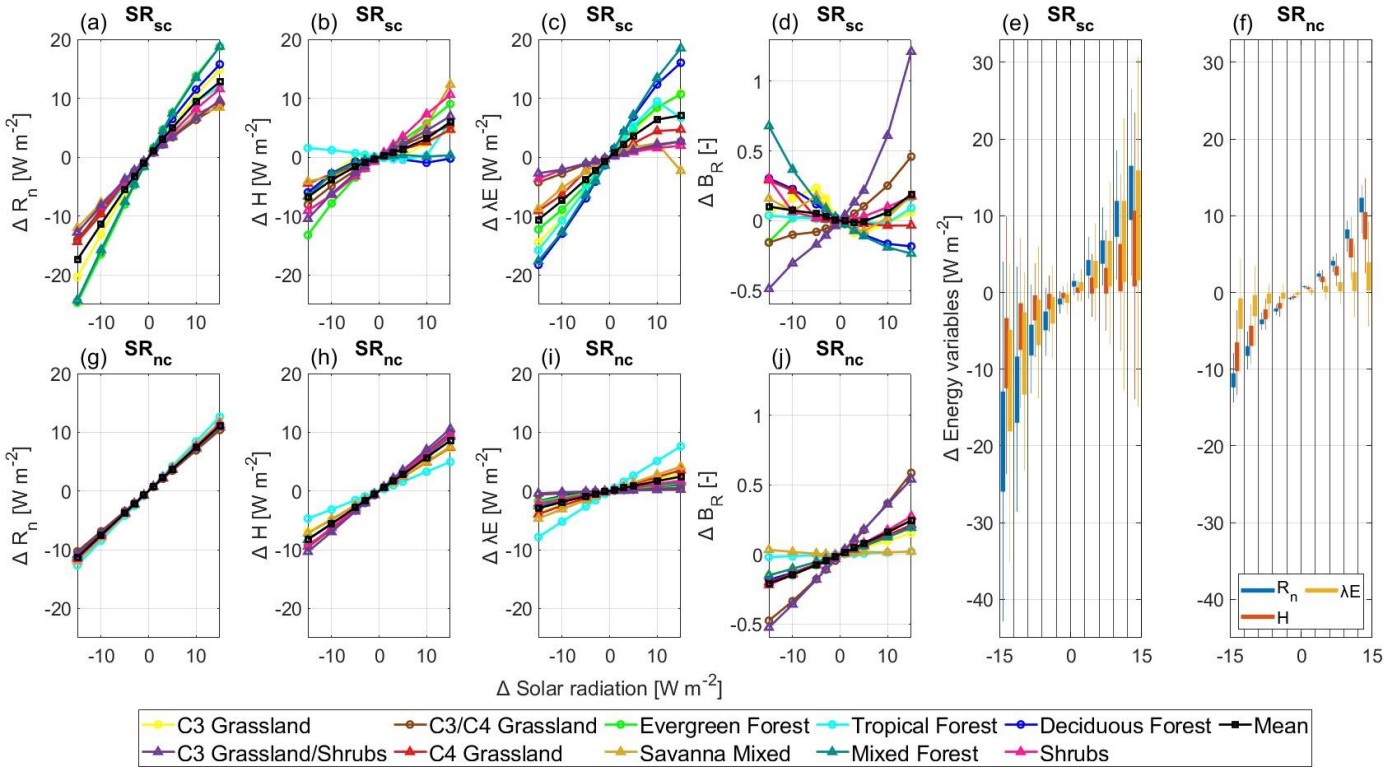

**Figure 4.** Changes in energy budget variables (a) (g) $R_n$, (b) (h) H, (c) (i) λE, and (d) (j) $B_R$ driven by surface solar radiation changes at 115 sites simulated with T&C under $SR_{sc}$ and $SR_{nc}$ scenarios. Coloured lines indicate changes in ten different biomes, and thick black lines indicate the average across all biomes. Boxplots (e) (f) represent the distributions of absolute changes [W m$^{-2}$] in $R_n$, H and λE under $SR_{sc}$ and $SR_{nc}$ scenarios, respectively. The cases where $R_{sw}$ change leads to ΔPR > ±

50% have been excluded as outliers.

### 3.3 Solar radiation changes – effects on the water budget

There is on average a positive correlation between changes in $R_{sw}$ and PR and ET in both $SR_{sc}$ and $SR_{nc}$ scenarios, while LK shows a negative response in the $SR_{sc}$ scenario and a positive one in the $SR_{nc}$, as a result of a larger positive sensitivity of PR to $R_{sw}$ changes, 0.015 mm day$^{-1}$ m$^2$ W$^{-1}$ in $SR_{nc}$ and 0.004 day$^{-1}$ m$^2$ W$^{-1}$ in $SR_{sc}$ (sensitivity computed as $R_{sw}$ changes from

345 -5 W m$^{-2}$ to 5 W m$^{-2}$, Table S4). The magnitude of hydrological changes in the $SR_{sc}$ scenario are generally greater than those in the $SR_{nc}$ scenario (Fig. 5d, 5e) despite the lower PR changes in the $SR_{sc}$ scenario (Fig. 5a, 5f), suggesting that changes in variables such as air temperature and vapor pressure deficit (Fig. S10) may impose a strong effect on the hydrological fluxes. The variations in runoff are relatively minimal (Fig. S11) compared to the other water fluxes (mean changes smaller than 0.05 mm day$^{-1}$ in both $SR_{nc}$ and $SR_{sc}$ scenarios). Surface runoff is not a considerable flux in the plot-scale ecohydrological

simulations (e.g., Fatichi et al., 2020) and hence, changes in PR mostly reflect in changes in ET and LK. Because of mass conservation (Eq. 2), the variations in ET and LK are of similar magnitude however with inverse sign when PR changes are modest as in the $SR_{sc}$ scenario, in which the median ET and LK changes are 12.2% and -8.9%, respectively, in the most extreme

+15W m$^{-2}$ R$_{sw}$ scenario (Fig. 5d). However, in the SR$_{nc}$ scenario, in which the magnitude of PR change is considerable (mean changes in PR is 0.15 mm day$^{-1}$ in the most extreme scenario with +15W m$^{-2}$ R$_{sw}$, Fig. 5f) and ET mean changes are less pronounced, i.e., 0.09 mm day$^{-1}$ under the SR$_{nc}$ scenario (Fig. 5g), there is a slight increase around 0.05 mm day$^{-1}$ in mean LK in the +15W m$^{-2}$ R$_{sw}$ scenario (Fig. 5h). In this case, the increase in PR more than compensates for higher ET, which is not the case in the SR$_{sc}$ scenario. The magnitude of PR change (mean change in PR is 0.04 mm day$^{-1}$ in the +15W m$^{-2}$ R$_{sw}$ scenario, Fig. 5a) is indeed less than that of change in ET (mean changes in ET is 0.25 mm day$^{-1}$, Fig. 5b). An increase/decrease in R$_{sw}$ leads to an ET increase/decrease in both scenarios and all biomes (except for the mixed savanna and tropical forest which start to show a decrease in ET from +10 W m$^{-2}$ to +15 W m$^{-2}$ R$_{sw}$ in the SR$_{sc}$ scenarios), but the magnitude of the increase is considerably higher in the SR$_{sc}$ scenario. Mean change in ET range from -0.38 mm day$^{-1}$ to 0.25 mm day$^{-1}$ in SR$_{sc}$ and from -0.10 mm day$^{-1}$ to 0.09 mm day$^{-1}$ in SR$_{nc}$ (Fig. 5b,5g) because the additional energy in the first scenario is transferred predominantly to $\lambda$E rather than H as discussed above. This is the result of a considerable increase in VPD and temperature in the SR$_{sc}$ scenario as R$_{sw}$ increases (Fig. S10). Without those changes, ET changes are much smaller. With higher Ta and VPD, vegetation tends to transpire more, which is the strongest driver of ET changes as ground evaporation and evaporation from interception do not change much (Fig. S12). Transpiration is also the driver of ET change in the SR$_{nc}$ scenario, but the magnitude of the change (from -0.05 mm day$^{-1}$ to 0.04 mm day$^{-1}$, Fig. S12f) is less than half that of the SR$_{sc}$ scenario (from -0.25 mm day$^{-1}$ to 0.19 mm day$^{-1}$, Fig. S12c).

While average changes are providing a summary picture of the effects of increasing solar radiation, PR, ET and LK show considerable differences in their trends for different biomes. Hydrological changes in C3 grasslands, deciduous forest and mixed forests were more pronounced in the SR$_{sc}$ scenario than in the SR$_{nc}$ scenario because these biomes in our analysis were mostly located at mid-high latitudes (Fig. S3), where temperature might be the most important factor rather than radiation influencing ET by limiting vegetation activity. In contrast, the hydrological variations in tropical forests are both remarkable in the SR$_{sc}$ and SR$_{nc}$ scenarios, suggesting that plants in the tropics are more dependent on radiation to alter hydrological fluxes through changes in photosynthesis and transpiration. It is worth noting though that savannas and tropical forests, both located in the tropics showed a turning point in their trends of ET and LK (Fig. 5b-c) at R$_{sw}$ changes above +10 W m$^{-2}$ in the SR$_{sc}$ scenario pointing to some form of water limitation induced by high radiation loads and temperatures. The detailed sensitivity information is presented in Fig. S13 and Table S4. The differences of sensitivity in regions characterized by different wetness index categories are rather minimal for SR$_{nc}$ (Fig. S14d-f, Fig. S15). However, the trends in PR (Fig. S14a) are different for the SR$_{sc}$ scenario in which dry sites experiencing lower precipitation and wet and intermediately wet sites showing higher precipitation with increasing R$_{sw}$. The magnitude of changes in hydrological variables show a larger increase in ET for wet sites with higher R$_{sw}$, and lower ET reduction in dry sites with a decrease in R$_{sw}$ (Fig. S15). These results are remarking the importance of water limitations in modulating the impacts of changes in R$_{sw}$ in the most extreme cases.

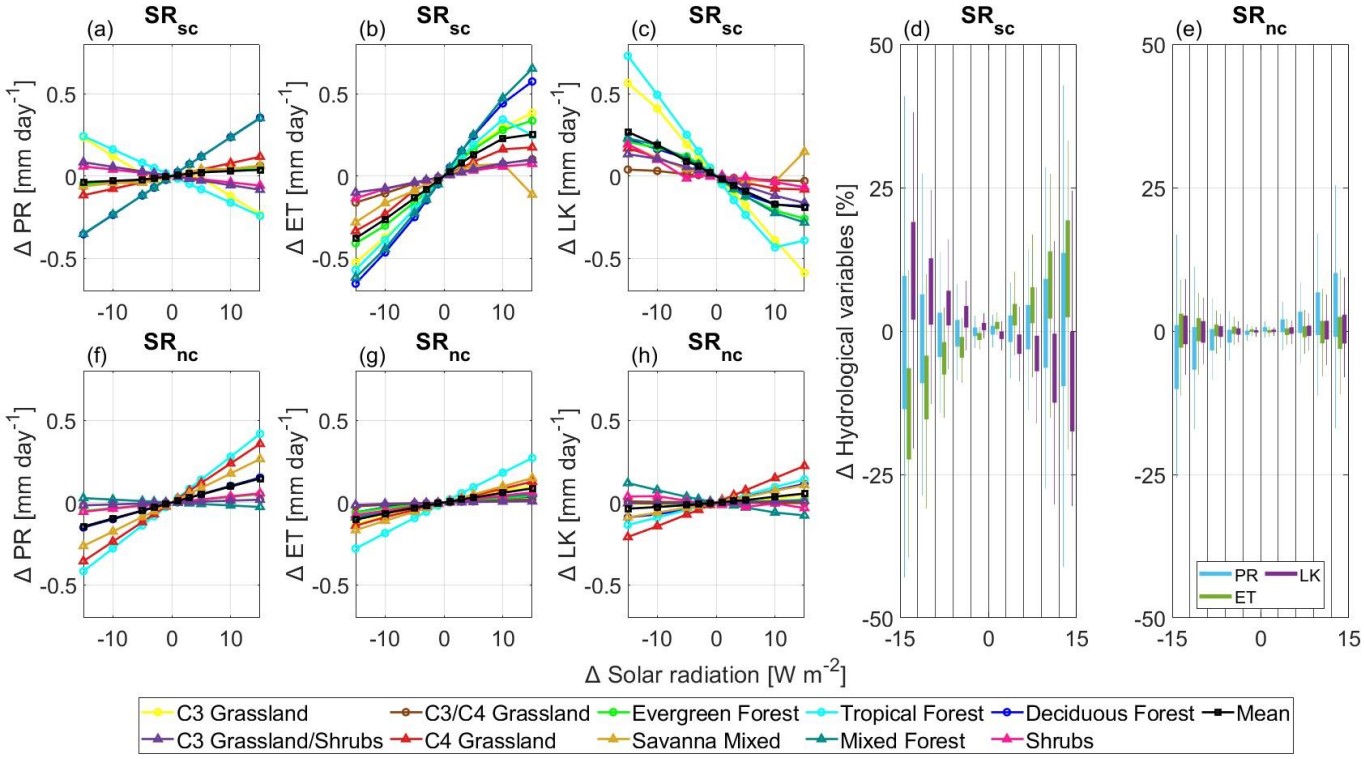

**Figure 5.** Changes in hydrological variables describing the water budget (a) (f) PR, (b) (g) ET, (c) (h) LK driven by surface solar radiation changes at 115 sites simulated with T&C under SR$_{sc}$ and SR$_{nc}$ scenarios. Colored lines indicate changes in ten different biomes, and thick black lines indicate the average across all biomes. Boxplots (d) (e) represent the distributions of relative changes [%] in PR, ET and LK in the SR$_{sc}$ and SR$_{nc}$ scenarios, respectively. To avoid non informative, high values, due to extremely low baseline ET and LK values, changes in ET and LK were rescaled based on their proportion of PR, for instance a 1% change in the plot is a 1% change on the ET/PR quantity. The cases where R$_{sw}$ change leads to ΔPR > ± 50% have been excluded as outliers.

### 3.4 Solar radiation changes – effects on vegetation productivity

As solar radiation increases, GPP changes nonlinearly in the SR$_{sc}$ scenario, which is different from the largely linear change in the SR$_{nc}$ scenario. The GPP changes in the SR$_{nc}$ scenario are of much smaller magnitude though than the GPP changes in the SR$_{sc}$ scenario, i.e., overall changes of -1.16 gC m$^{-2}$ day$^{-1}$ to 0.14 gC m$^{-2}$ day$^{-1}$ in SR$_{sc}$ and -0.21 gC m$^{-2}$ day$^{-1}$ to 0.13 gC m$^{-2}$ day$^{-1}$ in SR$_{nc}$ (Fig. 6). In SR$_{sc}$, the turning point from a slightly enhanced to reduced GPP occurs at or above a R$_{sw}$ change of around +5 W m$^{-2}$ (Fig. 6a). This is also the level at which B$_R$ starts to increase again (Fig. 4d), implying that beyond +5 W m$^{-2}$ of radiation change, the energy load combined with higher temperatures and VPD (Fig. S10) may move plants away from their optimal environmental conditions, and likely enhance water limitations in some locations, which causes a reduction in GPP (especially in Savannas and Tropical forest biomes, which are already warm environments). These results might also be affected by the fact that the T&C vegetation parameterization at each site is selected to reproduce local observations, and it might implicitly reflect some level of optimality in terms of radiation and temperature conditions, so that additional energy





and light are not beneficial. Conversely a decrease in $R_{sw}$ clearly reduces GPP considerably, especially in biomes located in temperate and cold regions (e.g., mixed forest, deciduous forest, and C3 grasslands, Fig. 6a, Fig. S3).

We use the $SR_{nc}$ scenario due to its linear GPP trend to compare the sensitivity of GPP to solar radiation in different biomes (Fig. S16) and found that C3/C4 grassland and C3 grassland/shrubs showed small negative sensitivities as those biomes are characterized by sparse vegetation and likely already light saturated, while the rest of the biomes showed positive sensitivities
to a change in solar radiation. Among them, the greatest increase in GPP was observed in the savanna areas and C4 grasslands, which are both ecosystems with higher amount of C4 photosynthesis, which has a lower intrinsic quantum efficiency (Singsaas et al., 2001), and thus is potentially benefitting more from additional light. The LAI response is in agreement with GPP changes, although the magnitude of the response varies from biomes to biomes (Fig. S17).

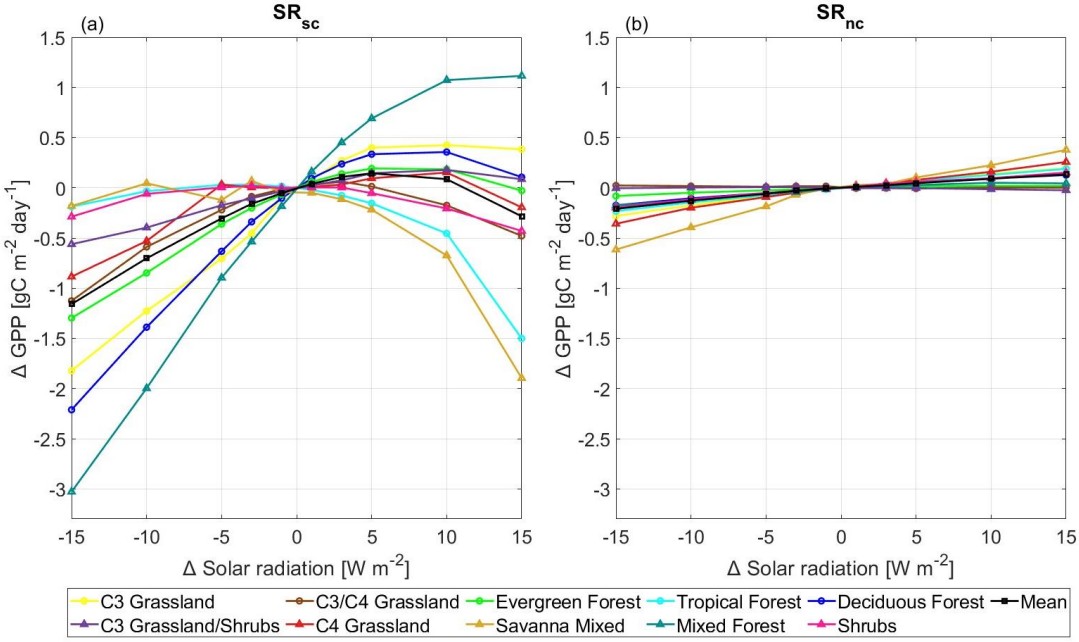

**Figure 6.** Changes in GPP driven by surface solar radiation changes at 115 sites simulated with T&C under the $SR_{sc}$ and $SR_{nc}$ scenarios. Coloured lines indicate changes in ten different biomes, and thick black lines indicate the average across biomes. The cases where $R_{sw}$ change leads to $\Delta PR > \pm 50\%$ have been excluded as outliers.

When we evaluated the response of different biomes to increased/decreased $R_{sw}$ in the $SR_{sc}$ and $SR_{nc}$ scenarios (Table S5), we
found that mixed forest and C3 grassland (most located in the mid-high latitudes) were the most sensitive biomes to changes in solar radiation under $SR_{sc}$ scenarios with the largest magnitude of GPP change, -19.3% and -15.0% with decreased $R_{sw}$ of 5 W m$^{-2}$ and +14.8% and +8.6% with increased $R_{sw}$, respectively. This is likely the result of increased growing season length in response to temperature. Shrubs were the least sensitive to decreases in light (0.1% GPP change), and C3 / C4 Grassland were the least sensitive to increases in light in the $SR_{sc}$ scenarios (0.2% GPP change), likely because water limitations are stronger
controls than temperature or light in these biomes.

## 4 Discussion

### 4.1 Solar radiation changes – energy and water flux responses

Since the 1960s, the world has experienced global dimming and brightening periods with trends in solar radiation shifting from a decrease to an increase with a turning point around the late 80s in the US and Europe (Wild et al., 2005). From 1961 to 1990, global surface solar radiation decreased by an average of 7 W m$^{-2}$ (about -0.2 W m$^{-2}$ per year) (Liepert, 2002) while from 1990 to 2005, surface clear-sky solar radiation increased at a rate of 0.66 W m$^{-2}$ per year (Wild et al., 2005). These changes in solar radiation and therefore energy incoming to the land surface are non-negligible and even larger changes could occur if solar radiation management due to geoengineering solutions is deployed in the future. Even though global scale studies have analyzed hydrological implications of geoengineering solutions (K. Ricke et al., 2023; Tilmes et al., 2013; Wei et al., 2018), it is still an open question how changes in surface solar radiation can affect the ecohydrological response of different biomes across the world. By determining sensitivities of climate variables to a change in solar radiation from CMIP6 experiments, we re-create two forcing scenarios that include (SR$_{sc}$) or exclude (SR$_{nc}$) the main land-atmosphere feedback of a solar radiation change at the land surface. We retrieved the known effects (Laakso et al., 2020) of precipitation scaling positively with radiation increase and evapotranspiration mostly following this pattern (Fig. 5). However, we also found that while R$_{sw}$ changes translate into R$_n$ changes almost unaffected by the presence of the land-atmosphere feedback (Table S6), the subsequent R$_n$ partitioning into H and λE is instead quite different when accounting for or excluding land-atmosphere feedback (Fig. 4, Fig. 7). When no feedback is included, the change in R$_n$ is mostly reflected in a change in H, with much less pronounced changes in ET and other hydrological variables (Fig. 5e, Fig. 7). However, once land-atmosphere feedback is included, which results in a change in temperature and VPD, the change in R$_n$ is more evenly partitioned into H and λE, with changes in ET/PR and LK/PR reaching up to ±20% in the most extreme R$_{sw}$ scenarios (Fig. 5d). In summary, for the same amount of R$_{sw}$ change accounting for land-atmosphere feedback promotes changes in ET and LK, even though changes in PR were more pronounced in the SR$_{nc}$ scenario (Fig. 5, Fig. 7). While we did not apply seasonally variable climate sensitivities to changes in solar radiation, our analysis suggests that summer season sensitivities, which primarily represents the vegetation growing season, shows the strongest correlation with the annual mean sensitivities (Fig S2). As most of the vegetation activities and ET occurs during summer months, our results should still be representative of the overall ecosystem response, especially as we considered a larger number of sites and thus a wide range of conditions where precipitation, air temperature and humidity change of different amounts in response to a solar radiation change (Fig. 3). However for ecosystems where winter hydrology might be very important to determine the growing seasons response of vegetation further analysis with seasonally variable sensitivities might be warranted.

### 4.2 Ecohydrological implications of an increase and decrease in solar radiation

As computed in this study, the ecohydrological response to R$_{sw}$ changes is influenced by a combination of energy partitioning, changes in hydrological processes, and vegetation response (Fig. 7). Here, we show that a change in R$_{sw}$ only, is unlikely to

have major implications on the hydrological and vegetation productivity as it mostly manifests in changes in H. This also implies that effects of global brightening on land-surface fluxes would not have been significant if global warming would not have concurrently occurred, and that observed trends in ET (Liu et al., 2021; Pan et al., 2020; Yang et al., 2023) in the 1980-2010 period are unlikely a direct consequence of changes in $R_{sw}$ alone. Furthermore, in the $SR_{nc}$ scenario, as $\lambda E$ and LK do not change much, the change in GPP tends to scale linearly with increasing light availability and is on average $\pm 0.2$ gC m$^{-2}$ day$^{-1}$ ($\pm 4.3\%$) for the most extreme $R_{sw}$ scenarios ($\pm 15$ W m$^{-2}$). Biomes with C4 plants (e.g. savannas) tend to be the most responsive, as the intrinsic quantum use efficiency of C4 plants is lower, while biomes with scattered and open vegetation (as C3/C4 grassland and C3 Grassland/Shrubs) have the mildest GPP response as they are likely already light saturated, and water limited. When land-atmosphere feedback is accounted for, a decrease in solar radiation is leading to a land-surface which is generally wetter, with lower ET and larger LK (and potentially streamflow once integrated at the catchment scale) and a considerably lower GPP. In the $SR_{sc}$ scenario, the GPP response to a negative $R_{sw}$ is much more pronounced with up to -1 gC m$^{-2}$ day$^{-1}$ (-21.4%) on average for a -15 W m$^{-2}$ $R_{sw}$ scenario. However, these changes are mostly caused by lower temperatures and VPD as the precipitation reduction is less pronounced in $SR_{sc}$. This shows that light or water limitations are not the main drivers of a negative change in GPP, but changes in temperature and VPD are. Therefore, it has to be expected that if solar geoengineering is deployed to counteract rising temperature levels, the overall hydrology and vegetation productivity will be much more similar to the present climate than shown in Fig. 5 and Fig. 6, as light reductions due to lower $R_{sw}$ in absence of temperature changes are less impactful than hypothesized accounting for less than 5% change of GPP even in the most extreme scenarios (Fig. 6b).

Conversely, in a scenario where changes in aerosols and cloud cover might lead to higher radiation loads, these will be accompanied by higher temperatures and VPDs, leading to significantly higher ET and reduced leakage, potentially jeopardizing water resources in certain regions. These conditions are sufficient to counteract the effect of higher light availability on GPP. GPP on average tends to peak at a $R_{sw}$ of +5 W m$^{-2}$ (even though variability across biomes is significant, Fig. 6a), and decreases at higher radiation loads because of higher temperatures and increased water limitations, reflected in a higher $B_R$. This suggests that vegetation, in the modelled locations, might be generally well adapted to current radiation and temperature conditions so that additional light availability does not stimulate GPP, with the exception of mixed forests, which are likely temperature limited in the current climate.

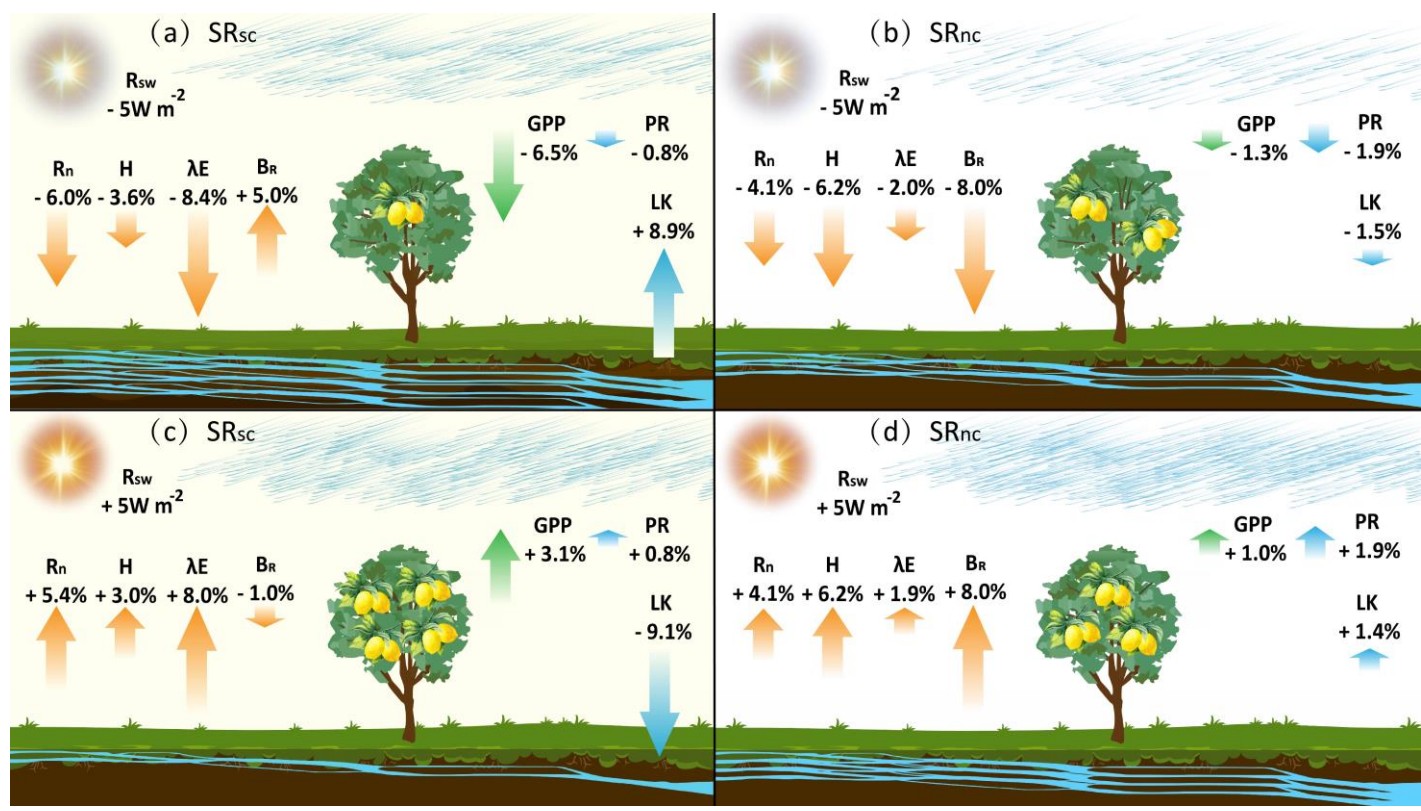

**Figure 7.** Ecohydrological response to (a) (b) decreased / (c) (d) increased $R_{sw}$ of ±5 W m$^{-2}$ under two different scenarios (a) (c) $SR_{sc}$ and (b) (d) $SR_{nc}$. The direction of the arrow represents the direction of the change, positive (upward) or negative (downward). Colors of the arrows indicate variables related to energy budget (orange), hydrology (blue), and vegetation gross primary productivity (green). The length of the arrows indicates the magnitude of change [%] compared to the control scenario.

## 5 Conclusions

We first quantified mean annual climate sensitivity to a change in solar radiation and further use these sensitivities to simulate ecohydrological responses induced by such a change in solar radiation accounting for or excluding land-atmosphere feedback in 115 sites around the globe spanning different biomes. The results show that a change in solar radiation itself modifies net radiation almost proportionally and led to substantially greater changes in H than λE with relatively minor implications for hydrology and vegetation productivity. The inclusion of land-atmosphere feedback caused by solar radiation changes led to a more pronounced change in $R_n$ and ecohydrological fluxes, with consequences also for vegetation productivity, especially when a radiation reduction is accompanied by lower temperatures. These results have implications for the re-assessment of global brightening and dimming effects on ecohydrological variables occurred in the past decades as well as on the evaluation of the potential changes in hydrological fluxes and vegetation productivity associated with solar radiation management projects.

**Code availability & Data availability**

Publicly available data was used in this study. CMIP6 model outputs can be obtained from https://esgf-node.llnl.gov/search/cmip6/. The T&C model code can be found at https://doi.org/10.24433/CO.0905087.v3

**Author contribution**

YW performed the data preparation, analysis of the results, prepared the figures and wrote a first draft of the manuscript. SF run the model simulations. NM and SF originated the idea and contributed to the writing.

**Competing interests**

The authors declare that they have no conflict of interest.

**Acknowledgment**

This study was supported by the National University of Singapore (Singapore Ministry of Education Academic Research Fund Tier 1) through the project "Bridging scales from below: The role of heterogeneities in the global water and carbon budgets" Award Number: 22-3637-A0001.

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
