# Peer review of "Ecohydrological responses to solar radiation changes"

_EGUsphere, 2024_

## Author Comment (AC1)

**#Reviewer 1**

This study employs a globally validated mechanistic ecohydrological model to simulate the ecohydrological responses to variations in solar radiation. A notable aspect of the author's approach is the avoidance of the simplistic method of merely adjusting incoming shortwave radiation. Instead, the study utilizes climate sensitivities derived from CMIP6 scenario simulations as inputs for the ecohydrological model. This innovative experimental design accounts for more realistic climate feedbacks.

Response:
Thank you for your positive evaluation of the manuscript, we will address the remaining comments as explained in the following.

Line 65, could you elaborate on how the term "pure" solar radiation change is defined? Given that the G1 simulation introduces some degree of land-atmospheric feedback at the local scale, it would be beneficial to clarify the meaning of "pure."

Response:
With "Pure" we referred to the difference between the control scenario and the G1 simulation because the G1 simulation increases $CO_2$ and decreases solar radiation, but maintains the temperature unaltered, which helps to isolate only the effects of changes in solar radiation. Hence, we call it 'pure' because it largely excludes the climate effect. However, as noticed by the reviewer "pure" is likely not the best term and we have modified it to "solar radiation changes in absence of large-scale temperature change".

Line 90 requires further explanation. Why is it impossible?

Response:
Compared to the short-term analysis (the second option), in the third option the feedback of the climate system to changes in solar radiation overlap with increased temperature, and the two effects are co-occurring, which makes them impossible to separate in the CMIP6 results without running other experiments. For our purpose, as we want to obtain a response to a change in solar radiation without confounding factors - as much as possible - we did not use these long-term sensitivities for the T&C simulations. This will be further clarified in the revised version.

Line 96 raises a question regarding the relevance of reporting climate sensitivities for the third case, as it seems tangential to the ecohydrological model. Conversely, why didn't you simulate the ecohydrological response to long-term climate feedback?

Response:
There have been other studies before looking at global-scale impacts of solar radiation changes on hydrology, the scope here was different. Our scope was first to look at what is the local/regional effect of changing solar radiation on the ecohydrology variables

without allowing the overall global climate to change, and second to provide a mechanistic interpretation and explanation for the observed changes from a land-surface perspective. For these two scopes, the T&C model was deemed adequate.
We did not include long-term T&C simulations because long-term sensitivities are affected by overall changes in climate dynamics, which tends to emphasize the global scale climate changes induced by an initial perturbation in solar radiation rather than the effect of solar radiation itself.

Line 101, does the G1 experiment solely modify surface solar radiation? I seek additional clarification on why most induced changes in climate variables are directly associated with radiation changes in this experiment (Line 117). What distinguishes G1 from the method mentioned in Line 80 (i.e., the first way)?

Response:
The G1 experiment consists of two main contrasting changes: an increase in $CO_2$ and a change in solar radiation to keep global temperature unaltered. Changes in $CO_2$ might have a minor effect on climate in this experiment, however since global temperature, which is the most closely related variable to express overall changes in climate (e.g., Seneviratne et al., 2016) remains constant, there is no feedback from a warmer or colder Earth, thus most of the induced changes in climate variables should be directly related to changes in solar radiation and to a minor extent $CO_2$. This is the experiment used in CMIP6 to isolate solar radiation effects, and the results are of high significance to build the no global climate feedback (SRnc) scenario. Please note that even though the changes in climate are induced mostly by solar radiation, other variables might still change a bit as illustrated in Fig. 1 in response to this solar radiation change. This will be clarified in the manuscript. By construction, using the "first way" will only modify Rsw without any change in any other variable.

Line 118 calls for an explanation of the model selection process.

Response:
We screened all the GCM models and found only the six models mentioned in the article providing results for the given experiments listed in Table 1. This will be clarified in the revised version.

In Table 1, it's noted that only one model (IPSL-CM6A-LR) is used to calculate short-term climate sensitivity and SRnc. When comparing ecohydrological responses between SRsc and SRnc, does this imply the exclusive use of this model? If not, how was the structural uncertainty among different models addressed? Moreover, including a line to detail the calculation of climate sensitivity based on the four experiments would be helpful.

Response:
Actually, we used four models (IPSL-CM6A-LR, CESM2-WACCM, CNRM-ESM2-1

and MIROC-ES2H) to compute the sensitivities for the SRnc scenario and three models (IPSL-CM6A-LR, MRI-ESM2-0 and CESM2) to compute the sensitivities for the SRsc scenarios. IPSL-CM6A-LR happened to have both the experiments for the SRnc (G1) and SRsc (abrupt-solm4p/abrupt-solp4p) scenarios. It is true that there will be uncertainty between the different CMIP6 models, this is accounted for by calculating the climate sensitivity based on the slope of the linear regression between the different models, so that uncertainty in the different models is smoothed. We will further clarify in the manuscript how we computed the climatic sensitivities, but basically these are the slopes of the linear regressions between changes in meteorological variables and surface solar radiation as obtained from the CMIP6 models."

Lines 132 and 133 necessitate further elaboration on the rationale for selecting the first decade and the last 50 years for computation.

Response:
The first decade was chosen as representative of a short-term response because it is expected that global temperature had not yet changed significantly over this period of time, at the same time we needed enough years to average internal climate variability and remove the uncertainty associated with the selection of one specific year. The last 50 years are instead characterized by a different global temperature which impact the overall Earth climate, so we chose this period as characteristic of "long term" climate effects induced by an initial change in solar radiation. The length of 50 years was also chosen to minimize the uncertainty associated with internal climate variability. We will further clarify these points in the manuscript.

Line 158, the term "no climate feedback" might be misleading since this scenario does involve feedback.

Response:
Thanks for the comment, we will modify this in the manuscript to "no temperature feedback".

Line 176, it would be interesting to know why leaf area index was not considered as a vegetation variable.

Response:
LAI is a prognostic variable in the T&C model and its changes have been simulated, however, they were not shown as the pattern of LAI is highly positively correlated with GPP. We have now included this analysis in the Supplementary Material.

There seem to be typographical errors at Lines 251 (fig. 3g?) and 254 (Fig. 3a?). Please verify the correct figure references.

Response:

Thanks for the comment. Yes, it is a typo and we will revise it in the revised version.

**References**

Seneviratne, S., Donat, M., Pitman, A. *et al.* Allowable $CO_2$ emissions based on regional and impact-related climate targets. *Nature* **529**, 477–483 (2016). https://doi.org/10.1038/nature16542

---

## Author Comment (AC2)

**#Reviewer 2**

The study develops a methodology to investigate the sensitivity of ecosystem response to changes in solar radiation, modified through geoengineering. This geoengineering link is very unclear both space and time scales-- whether or not actual geo-engineering implemented in a model, and what short- and long-term sensitivities are. The introduction should give a more thorough explanation of the problem, and the methods section should be more clear. Perhaps each of the four sensitivity approaches can be written in a separate paragraph.

Response:
Thanks for your comments. We will modify the introduction to provide further explanation of the link with geoengineering of the presented analysis. The four approaches to study how a change in solar radiation can modify ecohydrological variables are summarized at the beginning of section 2 and we will separate this section into different paragraphs to clearly illustrate what each approach entails. Essentially three simulation scenarios (the second/third/fourth approaches in the article) are using actual CMIP6 geoengineering simulations and we are using sensitivities of climate variables to these imposed changes in radiation to look at long-term climate feedback (abrupt-solm4p/abrupt-solp4p for the period January 1900 to December 1949), short-term climate feedback (abrupt-solm4p/abrupt-solp4p for the period from January 1850 to December 1859), and no temperature feedback (G1). Our aim is to compute changes in climate sensitivity due to changes in solar radiation that can be subsequently used to run ecohydrological simulations and thus study potential impacts of geoengineering projects, but more generally of changes in solar radiation, on the ecohydrological variables analyzed.

The results show that ecosystem response is relatively small, some fraction of the changes in ET, which is understandable. Although this sort of muted response is expected for the given sensitivities in the model forcing of 115 sitess, but I am unsettled with the way climate sensitivity is introduced based on the mean annual changes as reported in Figure 1. The future climate will bring a lot more variability across the globe, and that variability would potentially have a wider range of sensitivities than the sensitivity calculated using the global averages. In addition, sensitivities may also very more significantly when at least growing and non-growing season sensitivities are separated, as opposed to using sensitivities derived using global mean annual outputs. It is also unclear how climate sensitivities were implemented in each of the 115 sites. Are you doing something like bias correction in a time series obtained from a GCM somehow. Anyhow I detailed some of these issues below. Could be an interesting study after making all these issues clear.

Response:
I think Figure 1 might have generated a misunderstanding as it is indeed providing climate sensitivities at the global scale over land areas. However, the climate

sensitivities have been computed for each grid cell over the land area (Figure S3), including the 115 analyzed locations, and they have indeed a greater variability than the global average. In Figure 2 the red boxes show the global distribution of sensitivities when compared to the sensitivity distribution over the 115 sites and as you can see we sample most of the global variability using these sites. All sensitivities used for local-scale simulations are based on local values instead of global mean annual values. However, it is true that a single sensitivity in time is used without distinguishing for growing and non-growing seasons. We will test what is the difference in sensitivities across different seasons, to evaluate if this is indeed an important factor to consider.

There is no bias correction of the CMIP6 model outputs as climate sensitivities, e.g., how many mm day-1 the precipitation is changing per 1 W m$^{-2}$ of solar radiation change are then applied to the local meteorological observations of a given site that are available from the base case. So we just perturb the local observations. CMIP6 simulations are simply used to compute the local climate sensitivities using the closest pixel to the location of interest. We will clarify this methodological aspect in the manuscript.

More comments:

What is the scale of you modification of the solar radiation, local or global, because your simulations seem to focus on local hydrometeorology but when you are discussing those 4 scenarios you are mixing scales, local versus global and regional. Please give example of this solar radiation manipulation methods and their corresponding scales. At the end of the day I was not sure if any of your GCM runs actually incorporated ang geo-engineering methods. Or are you just grinding through the data to obtain delta Rn versus Delta other weather variables. The paper repeatedly uses short-term climate feedback and long-term climate feedback without describing neither. Is this like you implement a solar manipulation and look at the surface variables over and short and long durations later.

Response:
As answered in the previous response, in Section 3.1, we calculated and discussed the sensitivities at the global scale (global annual mean) to illustrate the concept, but we also calculated the sensitivities for each pixel and present the global sensitivity map in Figure S3. We applied the same linear regression method (used in Figure1) to each pixel of the 115 sites, then we calculated the slope of the linear regression and applied it as the climate sensitivity at each site. These climate sensitivities are then use to perturb the local meteorological observations. The CMIP6 simulations are built around geo-engineering methods that perturb solar radiation but our purpose is more generic: it is to understand climate sensitivities to a change in solar radiation (that can or cannot be driven by geoengineering) and the consequences of these changes on the ecohydrological variables. We will further clarify this point in the article.

The short-term climate feedback and long-term climate feedback are simply referring to how climate sensitivities are obtained as the short-term sensitivities (SRsc) were

computed over the first decade (Jan.1850-Dec.1859), and the long-term sensitivities SRlc were computed for the last 50 years (Jan.1900-Dec.1949). This will be clarified further in the manuscript.

Why would you want to separate the effects of radiation change on ecohydrology from everything else anyway, what is the type of GE this may be suitable for. If you only play with radiation, then that would be like a greenhouse experiment where one can manipulate radiation and keep temperature the way they wish, but there is no reason to test something that is not realistic. Besides, models are generally linear to Radiation, so the response can be predictable by some excel calculations.

Response:
We respectfully disagree that there is no value in separating the effect of a radiation change itself from the effect that a radiation change plus all the feedbacks on climate variables is generating. There is a theoretical argument that such separation can help explain which are the physical processes responsible for a change as illustrated in Figure 3 and 4 and associated discussion. There are also more practical implications as there could be conditions where changes in solar radiation might occur locally, for instance in response to an increase or decrease in aerosol concentration in a given region (brightening or dimming) but they do not have a global impact such as to modify the Earth mean climate. Additionally there are future geoengineering programs that aim to change solar radiation but at the same time keep temperature unaltered, and so in our case we need to understand what a pure solar radiation decrease does in order to understand the potential future ecohydrological response to a GE solutions. We agree that this point was not made clear in the previous manuscript version, but it will be included in the revised version.

Material and Methods lines 80 – 90:
Four options (O) were outlined for different geoengineering solution scales for manipulating radiation, but without what type of GE that might be, so that makes the alternatives hard to picture, below are my interpretations:

Response:
We apologize for ingenerating confusion. As written above our aim goes beyond implications of specific geoengineering programs and it is to understand climate sensitivities to a change in solar radiation (that can or cannot be driven by geoengineering) and the consequences of these changes on the ecohydrological variables. The four options are simply to progressively study changes that are solely driven by solar radiation or by solar radiation inducing short and long-term feedback on climate. This will be clarified further in the article.

O1. Just change radiation, all else being constant, suitable for local scale: I cannot picture what type of geoengineering solution this might, after all these GE solutions

target large areas.

Response:
This is indeed not a geoengineering solution but more the effect of a local change in solar radiation for instance because of changes in aerosol concentrations in a specific region.

O2. "..include short-term climate feedback, in which solar radiation changes lead to a modification of other climate variables.." Isn't this a fancy way of saying we can use a reginal climate model to downscale GCM outputs. Then the question is how you would implement GE within your RCM. But I don't this this was done at all.

Response:
It is not about how it is implemented, as the implementation is always the same: we applied climate sensitivities to local observations. It is more of what these climate sensitivities obtained from global scale simulations are representative of. This case is representative of a GE intervention in the early stage, where solar radiation changes immediately lead to some local feedback, e.g., locally warmer near surface temperature but there is no time to modify the overall Earth climate yet. We will clarify this further.

O3. " … all long-term climate impacts introduced by initial modification of solar radiation…" This can be practically done by modifying your local (let's say atmospheric chemistry) and feeding your RCM into the GCM. but how was this implemented?

Response:
See our previous reply, implementation is always the same, the values and meaning of the climate sensitivities differ though. Just to clarify, we do not run an RCM.

O4. ".. the fourth scenario is one in which solar radiation effects are isolated from global temperature changes by perturbing two variables, usually radiation is reduced, and $CO_2$ is increased to preserve the global scale mean temperature.." I am completely lost about the relevance of this.. what sort of geoengineering is this, what scale is this done, at the global scale we are already doing this experiment by putting more $CO_2$ in the atmosphere. Can you give more insights on this?

Response:
This is probably the most relevant scenario as geoengineering is designed to modify solar radiation and keep global temperature unmodified. In order to compute climate sensitivities for this case, we use the G1 experiment here and the perturbation conditions can be found on the CMIP6 website (please see: https://view.es-doc.org/index.html?renderMethod=id&project=cmip6&id=470998d5-c134-4684-b18b- ef551fb67293&version=1 ). We calculated local climate sensitivities and applied the local-scale sensitivities to the meteorological observations to run the T&C model.

The study ends up using O2 and O4 and looks at the impacts of this local hydrology, but also climate sensitivities of O3. I am not sure how this climate sensitivities of O3 was looked at, is the solar modification done globally, which model was this in the list? To put this in the language I understand, seems like O2 is essentially using GCM output in some fashion so that we can identify the ecosystem response to Delta_Solar.

Response:
We hope that the answers to the above questions have clarified how the methodology has been implemented. We apologize again for the confusion we have ingenerated. The revised version will explicitly mention that we applied the climate sensitivities to local observed meteorological variables.

Lines 95-100: are those hydromet-sensitivities calculated from GCMs from the locations of those 115 sites? It would be very tricky to downscales from GCMs to those locations, how was this done? I presume those different CO2 levels were also included in the simulations, or maybe not, it depends what these Geoengineering scenarios are. Without more guidance on geoengineering solutions, it is difficult to judge the study.

Response:
We obtained the sensitivities for each grid cell (Figure S3). The 115 local sensitivity was selected for the closest pixel to the location of interest, however, given the fact that sensitivity have a smooth geographical variability – see Figure S3, sensitivities are the same in the range of several kilometers. It was not tricky to downscale as we simply applied the climate sensitivities to local observed meteorological variables, so we only rely on GCMs for the computation of sensitivities not for the simulation of local climate conditions.
While some of the geoengineering experiment increased the $CO_2$, we didn't perturb the $CO_2$ level in the T&C simulations. This is because we want to understand the effect of solar radiation changes not the overall consequences of a specific geoengineering experiment. This will be clarified and explained better in the revised manuscript.

Lines 125: I see that the sensitivity is actually calculated using the annual values. Is there evidence that these relationships obtained from annual values represent the seasonal changes.   shouldn't this be done perhaps using a moving-average that cover the seasonal trends between solar-precip-temperature etc., especially considering the fact that you have an hourly model.   Or if not, how do you justify? I presume your ecohydrology model response would be most sensitive to the "growing season" however it is naturally different across the 115 locations. Again here, I would like to go back and think about how the initial downscaling was made, that also would have major implications on the local weather variables, perhaps not so much the solar.

Response:
It is true that a single sensitivity in time was used without distinguishing for growing

and non-growing season or more generally for different seasons. We will test what is the difference in sensitivities across different seasons, to evaluate if this is indeed an important factor to consider.

Lines 192-195: what is the basis and evidence for this, your analysis is global and I am not seeing any citations to support for this claim.

Response:
We concluded this from Figure S4. There are a few isolated instance where surface solar radiation can have a positive change in response to a decrease in top-of-the-atmosphere solar radiation and a negative change in response to the top-of-the-atmosphere solar radiation. These are clearly driven by atmospheric dynamics and changes in cloud patterns. We will cite Figure S4 in the revision.

Figure 1. As expected annual trends seems fairly muted.

Response:
It is true that the global annual average sensitivity is not significant, but here we only present the global sensitivity. For the actual simulations we use local sensitivities as illustrated in Fig. 2 and discussed above.

Line 225. How did you compute the climate sensitivities at those 115 locations, again wouldn't this require some sort of GCM downscaling?

Response:
Please see our replies to the above questions.

---

## Author Response (AR2)

**#Editor**

We have now received new comments from the two referees on the revised paper. I agree with the reviewers that the manuscript presents a valuable contribution, but there are still comments to be addressed (comments by reviewer #2) before considering publication, including:

1) Inclusion of a figure (flow chart or conceptual diagram) to better explain the methodological approach.

Response:

Thank you for your valuable comments. We have added a conceptual diagram (Figure 1) illustrating the workflow of our research, including how we use the data and methodology.

2) Clarification on model testing.

Response:

Thank you for your comments. The T&C model has been validated in numerous previous studies. We have explained this better and included references to relevant validation studies in Section 2.3.

3) Addressing the comment on seasonality effects (It would be good to add some comments/paragraph in the discussion, mentioning possible effects). I believe these are all very useful comments and feedback that need to be answered/addressed to further improve the manuscript.

Response:

Thank you for your comments. We added some seasonal analysis in section 2.2 and in the discussion part, specifically at the end of Section 4.2. Additionally, we have highlighted the limitations of this study and outlined potential directions for future research.

**#Reviewer1**

None

**#Reviewer2**

In the response document, the authors frequently use future tense as in this example – "… We will test what is the difference in sensitivities across different seasons, to evaluate if this is indeed an important factor to consider.." Please state more clearly and indicate whether or not you tested or made actual changes in response to a comment, and please include line numbers. Apparently the seasonality comment I had was not adequately addressed by providing results.

Response:

Thank you for your comments. We have computed seasonal sensitivities across winter (December-January-February: DJF), spring (March-April-May: MAM), summer (June-

July-August: JJA), and autumn (September-October-November: SON) and annual mean sensitivities, for precipitation (PR), temperature (TA), specific humidity (SH) and wind speed (WS) changes in response to a solar radiation changes for the 115 sites in Figure S1 and Figure S2, and we have added some discussion about seasonality analysis in section 2.2 line 158-162 in tracked version. In this round revision, we added further details about the seasonality analysis and potential limitations in discussion part.

I now understand that geo-engineering angle in the paper is not the necessary drive, copying from the response—".. but our purpose is more generic: it is to understand climate sensitivities to a change in solar radiation (that can or cannot be driven by geoengineering).."I still cannot fully wrap my head around your work, and cannot reproduce it from the text. I would love to see the authors to develop a figure (flow chart, perhaps with drawings and conceptual diagrams) such that I can follow through the logic algorithmically and see exactly what they did and how I can reproduce it.
Response:
Thank you for your comments. We have added a conceptual diagram (Fig.1) illustrating the workflow of our research, including the data and methodology. We hope this addition is helpful.

My comments on seasonality are not addressed. I was just curious how growing and dormant season effects on water balance components would look like. I suspect there are some snow in some of these sites, perhaps the GPP may not be all that sensitive to this seasonal separation, but the reader would be curious to see how solar radiation be different on water balance components and whether GPP is actually has any strong connection to changes in dormant season water balance.
Response:
Thank you for your comments. As sensitivities at annual scale are best correlated with sensitivities during the growing season (Fig S2), we think that our climatic perturbations are representatives of the main changes in climatic variables that affect the functioning and response of vegetation. Plus given the numerous sites and the different sensitivities in each of those, in the presented results we are already testing a wide range of conditions where precipitation, air temperature and humidity might change of different amounts in response to a solar radiation change. In other words, the final results in Fig 4, 5 and 6 are unlikely to be affected by the exact values of the sensitivities, as far their magnitude and overall direction is correct. However, as we do not force the ecohydrological T&C model using seasonally variable sensitivities, we are remarking this as a limit in the discussion section (Section 4.2).